# Strategies for the Mobilization and Deployment of Local Low-Value, Heterogeneous Biomass Resources for a Circular Bioeconomy

**Fabian Schipfer [1,*](ID), Alexandra Pfeiffer [2](ID) and Ric Hoefnagels [3]**

1 Energy Economics Group, Technische Universität Wien, Gusshausstraße 25-29, 1040 Vienna, Austria
2 Deutsches Biomasseforschungszentrum, 04347 Leipzig, Germany; alexandra.pfeiffer@dbfz.de
3 Copernicus Institute of Sustainable Development, Universiteit Utrecht, 3584 CB Utrecht, The Netherlands; r.hoefnagels@uu.nl
* Correspondence: schipfer@eeg.tuwien.ac.at

**Abstract:** With the Bioeconomy Strategy, Europe aims to strengthen and boost biobased sectors. Therefore, investments in and markets of biobased value chains have to be unlocked and local bioeconomies across Europe have to be deployed. Compliance with environmental and social sustainability goals is on top of the agenda. The current biomass provision structures are unfit to take on the diversity of biomass residues and their respective supply chains and cannot ensure the sustainability of feedstock supply in an ecological, social and economical fashion. Therefore, we have to address the research question on feasible strategies for mobilizing and deploying local, low-value and heterogeneous biomass resources. We are building upon the work of the IEA Bioenergy Task40 scientists and their expertise on international bioenergy trade and the current provision of bioenergy and cluster mobilization measures into three assessment levels; the legislative framework, technological innovation and market creation. The challenges and opportunity of the three assessment levels point towards a common denominator: The quantification of the systemic value of strengthening the potentially last remaining primary economic sectors, forestry, agriculture and aquaculture, is missing. With the eroding importance of other primary economic sectors, including fossil fuel extraction and minerals mining, the time is now to assess and act upon the value of the supply-side of a circular bioeconomy. This value includes the support the Bioeconomy can provide to structurally vulnerable regions by creating meaningful jobs and activities in and strengthening the resource democratic significance of rural areas.

**Keywords:** bioeconomy strategy; regional development; residues; policy; market; technology; commoditization

## 1. Introduction

The European Bioeconomy Strategy [1] aims to "strengthen and boost biobased sectors". By definition, the bioeconomy includes "all primary production sectors that use and produce biological resources (agriculture, forestry, fisheries and aquaculture); and all economic and industrial sectors that use biological resources and processes to produce food, feed, bio-based products, energy and services. To be successful, the European bioeconomy needs to have sustainability and circularity at its heart."

Since the 1970s, industrial ecology and industrial metabolism discussions coin the term Circular Economy (C.E.), the C.E. has been used as a guideline in policy-making, especially in China and Europe. Today the C.E. is mainly attributed to electronic waste (see Circular Economy Action plan [2]) and recently also plastics (see Plastic Strategy [3]).

The Annex of the draft proposal for a European Partnership for a Circular Biobased Europe [4] argues why also a bioeconomy is inherently a C.E.; biobased sectors have $CO_2$-avoidance and retention, reduction, recycling and reuse of wastes and residues as its

goals, all traits which have been primarily credited to the circular economy. The success of both the circular bioeconomy and the broader circular economy depend on a sustainable feedstock supply. However, shifting the respective primary economic sectors, i.e., the feedstock supply, to sustainable practices comes with considerable technical, societal and organizational challenges that have to be addressed [5,6].

The current bioenergy provision is mainly based on wood chips, wood pellets or first generation biofuel plantations [7]. The underlying resources are mobilized primarily for material services (e.g., construction wood, pulp and paper) [8], while first generation biofuel resource provision use similar production techniques and supply chains as agricultural production. As a result and in their review on bioenergy supply and demand scenarios and projections, Mandley et al. [9] stress a potential mismatch due to limited modelling and analysis of crucial conversion processes between fresh biomass and end-user services. A plethora of underutilized, non-commodity biomass resources is still not touched upon, which could become the feedstock basis for the circular bioeconomy of tomorrow. These resources can be categorized in energy crops-, forestry and agricultural residues, and biogenic waste [10–13]. However, they are diverse in, e.g., physical properties (energy density, moisture content, ash content but also contamination such as sand/plastic), origin (landscape management, residential garden/kitchen waste) and legal status (waste vs. resources/material). The current biomass provision structures are unfit to take on this diversity and cannot ensure the sustainability of feedstock supply in an ecological, social and economical fashion. Therefore, we have to address the research question on feasible strategies for mobilizing and deploying these local, low-value and heterogeneous biomass resources.

Thus, and for the present paper, we are building upon the work of the IEA Bioenergy Task40 scientists and their expertise on international bioenergy trade and the current provision of bioenergy. To address all sustainability dimensions, mobilization strategies have to respect planetary boundaries [12] and have to be financially viable and contribute to other societal goals. Especially for the provision of local and low-value biomass resources, this means supporting structurally weak and rural regions. The research focuses on the European Union concerning policies but is also inspired by technology- and market developments in the rest of the world.

## 2. Materials and Methods

This work is based on an extensive discussion on biomass mobilization strategies between International Energy Agency (IEA) Bioenergy Technology Collaboration Program (TCP) Task 40 scientists. The expertise of the authors and discussion participants undoubtedly defines the scope of the presented findings. Task40 initially focused on international bioenergy trade. However, the established supply-chain knowledge proved to be applicable to strategic questions about biomaterials as well (see, e.g., Schipfer et al. [14]). The international consortium specializes by now on the "deployment of biobased value chains" in support of a broader, circular bioeconomy. Systemic assessments, including the utilization of bioenergy, as, e.g., discussed for energy system models in Chang et al. [15], are increasing in spatial, temporal and sectoral resolutions. The IEA Bioenergy Task40 follows this zeitgeist by dedicating a task force to *Regional Transitions* studies. For the sake of this paper, we understand "region" as an area that could have its own characteristics or even administration. We refrain from setting a precise definition, but as a rule of thumb, "regions", "regional" and "local" could span from municipalities, the lowest local administrative unit to groups of districts, or the NUTS 3 level.

For this paper, the IEA Bioenergy Task40 experts focused on transferring and extending their knowledge on current bioenergy carrier provision structures to the local, low-value feedstock base of tomorrows circular bioeconomy. During the discussions within the Task force and based on previous works on mobilization strategies for bioenergy of lower spatial and sectoral resolution (e.g., Junginger et al. [7]), we collect information on respective current developments, barriers and opportunities. The discussion is further complemented

by scientific literature on the identified topics and a collection of unpublished research- and development projects. The here presented collection does not claim completeness or indicates any ranking of importance. Instead, it aims at creating a coherent reference work on challenges and opportunities for novel biomass provision structures. It should be used to derive key concepts for follow-up scientific-, market- or patent research. To facilitate the analysis and discussion beyond the project, we cluster the topics into three categories; legislative framework, market structures and technological innovation (see Figure 1).

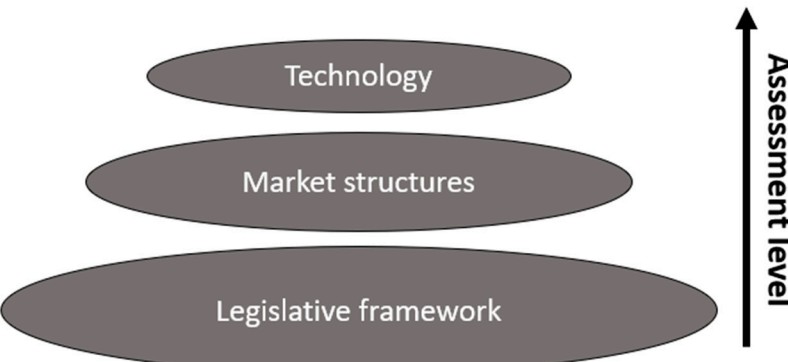

**Figure 1.** Biomass mobilization strategy categories. The arrow points in the direction of increasing assessment resolution. Source: own illustration.

This paper's Results and Discussion (Section 3) are structured following the outlined categories, starting with the lowest assessment level, highlighting top-down the current developments, opportunities, and barriers in the European legislative framework before zooming into the highest assessment level on bottom-up technological innovation mobilization strategies. The Results and Discussion section is completed with an analysis of biomass markets for energy and material use. The Conclusions section (Section 4) connects the different assessment levels back together and provides recommendations and limitations of the present study.

### 3. Results and Discussion

#### 3.1. Legislative Framework for Biomass Mobilization

For the basic assessment level of biomass mobilization strategies, we focus on the European Union and its common legislative framework for the 27 Member States (M.S.). We first and foremost are interested in high-level documents labelled "strategy", "blueprint", "roadmap", or "action plan". Even though these terms lack clear definitions and unambiguity, they are often used sequentially (1) with a strategy outlining a general perspective for development, while (2) blueprints and roadmaps are frequently used to illustrate one or several development paths and timelines. (3) Action plans should include concrete steps or action points and preferably quantifiable goals. On an E.U. level, respective documents facilitate the non-juridical discussion of overall trends, even though outlined targets and measures are of non-binding characters eventually to be implemented in regulations and directives. E.U. regulations are binding in their entirety in all Member States/M.S., while directives are to be "transposed" into national laws of the M.S. In contrast, decisions (addressing particular States or organizations) and ordinances on a national, regional or sub-regional level are out of the scope of the present paper.

In the following sub-sections, we provide a top-down mapping of the legislative frameworks of high relevance for mobilizing local, low-value and heterogenous biomass. We explore the E.U. policy landscape (Section 3.1.1), international projects on regional mobilization strategies (Section 3.1.2) and how the novel concept of Multilevel governance tries to bridge local with E.U. governance (Section 3.1.3).

### 3.1.1. EU Policy Environment Affecting Regional Biomass Mobilisation

The European Green Deal lays down the strategy for a broad set of E.U. policies currently formulated and enacted between 2019 and 2024, building upon the existing policy framework [16]. To achieve climate neutrality by 2050, the Climate Target Plan proposes to cut greenhouse gas (GHG) emissions by 55% in 2030 through a combination of legislation on the Emission Trading System (ETS), Effort Sharing, and Land Use [17]. The current proposal on revising the Renewable Energy Directive includes the amendment of renewable energy targets to 40% by 2030, quantitative sector-specific renewable energy goals for buildings, transport, industry and district heating and the tightening of sustainability criteria for biomass [18]. Biodiversity protection in forests, GHG saving criteria for existing bioenergy installations (as small as 5 MWel), phase-out of electricity-only production from biomass, and enforced cascading principles are proposed by the European Commission.

Coherence with already existing and to-be-revised documents has to be ensured. The primary strategy (or action plan) on the European level for biomass mobilization can be seen in the updated E.U. bioeconomy strategy [1]. This document complements similar objectives to the 2012 bioeconomy strategy with main action areas, including deploying local bioeconomies rapidly across Europe. A Strategic Deployment Agenda (SDA) "for sustainable food and farming systems, forestry and bio-based production in a circular bioeconomy" was envisaged to be finalized by 2021 [1]. This "roadmap" will optimize "synergies between the Common Agricultural Policy (CAP), [maritime and] fisheries [policies], [the] Agricultural Fund for Rural Development (EAFRD), other European Structural and Investment Funds (ESIF)" and mobilize the agricultural European Innovation Partnership (EIP-AGRI). Furthermore, the Covid pandemic brought the relevance of healthy regions off the urban centers to the fore. Considerable increases in grants and loans are proposed in the NextGenerationEU package to be directed to recovery measures but also to rural development [19].

Furthermore, local bioeconomy development is supported for coastal (e.g., Blue Bioeconomy grants), urban (Urban Circular Bieoconomy Strategy funding) and rural areas (in national CAP strategic plans). The strategy also aims at piloting carbon farming initiatives to "make carbon sequestration and emission reduction a profitable farming/forestry activity". Finally, E.U. Bioeconomy policy support facilities (via the BIOEAST initiative) and a European Bioeconomy Forum for M.S. is initiated. Furthermore, and under the Green Deal, the Just Transition Mechanism (JTM), including the Just Transition Fund (JTF), the InvestEU "Just Transition" scheme, and the European Investment Bank (EIB) public sector loan facility will provide support to "reduce regional disparaties and to address structural changes in the E.U." related to the transition towards climate neutrality. In addition, the public–private partnership for circular bioeconomy R&D&D (now CBE JU, former bio-based industries joint undertaking—BBI JU) with potential impacts on biomass mobilization has to be mentioned as an essential tool to foster innovation in regions to mobilize biomass for a circular bioeconomy.

Focusing on regional development, the E.U. Long-term Vision for Rural Areas aims at strengthening the provision of "food, homes, jobs, and essential ecosystems services". Proposed measures include a rural revitalization platform, R&I for rural communities, boosting digital connectivity and competencies, establishing carbon-sink focus areas and fostering rural entrepreneurship [20]. The Farm to Fork Strategy addresses sustainability throughout the life-cycle of our nutrient services, including "production, processing/distribution, consumption [and] food loss and waste" prevention [21]. The Forest Strategy for 2030 promotes a sustainable forest bioeconomy, including the use of wood-based resources but also eco-tourism while "ensuring forest restoration and reinforced sustainable forest management for climate adaptation and forest resilience" [22]. The Biodiversity Strategy for 2030 aims at the same time to establish and extend an "EU-wide network of protected areas on land and at sea" and announce "binding nature restoration targets" [23].

More specifically, and based on the current CAP (2014–2020), farmers have to set aside a mandatory share of 5% of farmland for Ecological Focus Areas (EFAs), including

"grasslands, hedges, buffer strips or nitrogen-fixing crops" [24]. Short rotation plantations (SRP), including short rotation coppice (SRC) and single-stemmed trees (SRF), count towards EFAs. However, implementation of these greening measures is still limited to about 50.000 hectares in Europe, with considerable shares in Sweden due to an unrelated willow plantation trend between 1986–1996 and some measures such as establishment grants in the U.K., Ireland and Germany [25].

This section provides a preliminary list and description of E.U. policies that should be considered when planning for the mobilization of local, low-value and heterogenous biomass feedstock. Strategically deploying these top-down resources is subject to regional, national and international efforts. The next section addresses the evolution of respective projects on regional biomass mobilization.

### 3.1.2. Regional Strategies Focusing on Regional Biomass Mobilization

Historically, the E.U. Biomass Action Plan from 2005 aimed at setting out "measures to increase the development of biomass energy from wood, wastes and agricultural crops by creating market-based incentives" [26]. To promote regional structures, the E.U. Biomass Action Plan was followed up by several regional biomass action plans, bioenergy action plans and regional development plans. These plans are often developed for specific regions, countries, and sometimes inter-regional partnerships by trans-disciplinary project consortia, including research and regional energy agencies, biomass associations, and regional stakeholders. Renown projects, their funding source and runtimes are listed in Table 1.

**Table 1.** Projects aiming at creating and implementing regional biomass action plans with multi-regional and international scopes. Source: own elaboration.

| Funding Agency | Project Name | Project Years |
|---|---|---|
| IEE [1] | REGBIE+ | 2007–2009 |
| EFRD [2] | 4Biomass.eu | 2009–2011 |
| IEE [1] | Bioregions.eu | 2010–2013 |
| Interreg [3] | Bio-En-Area | 2011–2015 |
| IEE [1] | Biomass Policies | 2013–2016 |
| FP7 [4] | S2Biom | 2013–2016 |
| IEE [1] | Basis | 2013–2016 |
| Horizon2020 [5] | BioVill | 2016–2019 |
| Interreg [3] | Bio4Eco | 2016–2020 |
| Nordic Council of Ministers | Nordic Bioeconomy Prgrm | 2018–2022 |
| Horizon2020 [5] | BioEastsUp (initiative) | 2019–2022 |

[1] Intelligent Energy Europe, [2] European Funds for Regional Development, [3] Innovation & Environment Regions of Europe Sharing Solutions, [4] 7th Framework Program for Research of the European Commission, [5] Horizon2020 Funding Program of the European Commission.

The selection of the projects listed in Table 1 is based on their multi-regional and international scope. Project consortia consists of partners from 6–13 Member States, including neighboring countries. Following up on the E.U. Biomass Action plan projects until 2013 mainly focused on creating action plans and action plan templates and improving regional policies for bioenergy uptake and market creation. Older, potentially fitting projects such as "Make-it-BE", "BioMob", "BioCLUS" and "Rok-FOR" are mentioned in the review on regional biomass planning by Kautto and Peck [27]. Still, information on these projects is insufficient for further analysis. Since 2013, sustainable and efficient use of biomass and the interaction between biomaterial, food and feed and bioenergy based on supply chain approaches is in the foreground. For this purpose, especially the "Biomass Policies" and the "S2Biom" projects mapped sustainable supply potentials. They are published as openly accessible and "updated harmonized datasets at a local, regional, national and pan-European level for EU28, Western Balkans, Moldova, Turkey and Ukraine" [28]. More recent projects focus on establishing "knowledge-bio hubs" and "bio villages", exchanging

information and know-how between regions. The Nordic Council of Ministers representing Denmark, Finland, Iceland, Norway, Sweden, the Faroe Islands, Greenland and Åland furthermore established the "Nordic Bioeconomy Programme" to collaboratively improve the use of biogenic residues and to remain in a leading position with regard to regional bioeconomy development.

European regions concerned about structural losses in the primary economic sectors through the fossil fuel phase-out also show interest in strengthening local and regional bioeconomies. These regions include mainly coal mining since self-sufficiency rates of the E.U. are at 60% for coal, 16% for natural gas and only 5% for oil and petroleum products (in 2019) [29]. The Horizon 2020 project TRACER supports regions in Bulgaria, Germany, Greece, Czech Republic, Poland, Romania, Serbia, Ukraine and the United Kingdom in designing or re-designing their Research and Innovation strategies. The project "analyses the impact of the energy transition, in terms of social change, communities shrinking, migration, demographic ageing, poverty, high youth unemployment rates and participation to education and training" in intensive coal regions [30]. The mono-industrial character of these regions makes them specifically vulnerable to respective socioeconomic challenges and asks for dedicated measures, especially regarding re-skilling, job creation, a productive re-usage of the industrial landscape, investments in infrastructure and addressing legal issues related to land ownership. Shifting the focus from primary economic sectors to secondary or tertiary sectors might not be feasible for some of these regions; thus, coupling fossil-fuel phase-out with bioeocomy actions are promising strategies. A regional bioeconomy transition example is reported for the chemistry park "Schwarzheide" with leading research parties developing and pioneering in the deployment, together with BASF company and other plastics processing companies, of bioplastics and biodegradable synthetics [31].

The strengthening of primary economic sectors in these regions will have to focus on sustainable management of agriculture and forestry for a circular bioeconomy. The circular bioeconomy sectors will cover multiple services, including electricity, heat, chemicals, bio-based materials, food/feed and services on ecosystems and the carbon budget. Respective bioeconomy concepts stand in stark contrast to a direct substitution of coal with biomass for electricity-only at the same scales of incumbent coal-fired power plants. Subsidies for bioelectricity and electricity-only are already phased out, e.g., in the Netherlands to re-orientate limited biomass resource potentials to economic sectors, which are more challenging to abate [32]. An obligatory phase-out is proposed by the European Commission for all Member States, starting with 2026, except for Bioenergy Carbon Capture and Storage (BECCS) or plants located in a "region identified in a territorial just transition plan" [33].

In summary, we can highlight that numerous regional development and biomass action plans have been developed in dedicated projects over the last decades. The focus and scope of these plans co-evolve with the policy framework. In the last decade, the main driver was the provision of first generation biomass feedstocks for bioenergy and meeting renewable energy targets. Since then, the aim of these plans seems to have shifted to (1) more generic approaches, including quantitative feedstock and market potential assessments (2) with a more holistic view on different bioeconomy sectors. Regions dependent on coal mining now have the opportunity to tap into this evolution of action plans. Creating added value and opportunities for structurally weak and vulnerable regions has been an objective in the projects following up upon the E.U. biomass action plan of 2005. However, and with the enlargement of the E.U., barriers and opportunities multiplied that have to be eventually addressed based on multilevel governance frameworks.

### 3.1.3. Multilevel Governance for Biomass Mobilization

The E.U. is a valuable case study for biomass mobilization strategies due to its great variety in biomes, economic structures, governmental forms and cultures but with the ambition to agree on the direction forward without inhibiting the diversity in approaches for the actual progress. It is not surprising that the member states themselves have a similar

governance philosophy, resulting in a highly dynamic patchwork of legislative structures. The Multilevel Governance (MLG) concept provides a framework for acknowledging the interactions between the different spatial or organizational resolutions while giving room for a more coherent policy mix across these resolutions and across sectors. An MLG view is essential, especially for enabling local and regional energy and climate initiatives. Dobravec et al. [34], for example, "analyses the existing energy planning governance in Austria throughout the MLG-structure by focusing on the alignment between the local energy and climate initiatives and the national and E.U. goals". They find that a "general willingness of Austrian municipalities to take part in local energy actions" as well as "cooperation of different levels of governance from the top-down and bottom-up perspective" via local, regional and inter-regional initiatives such as the e5—Program of energy-efficient municipalities, KEM—Klima- und Energiemodellregionen, CoM—Covenant of Mayors can be observed. Based on the identified shortcomings, especially concerning data availability and spatial energy planning for renewables crossing different jurisdictions and responsibilities, the paper recommends extending the existing governance on multiple levels with a more flexible MLG, including neighborhoods and zones and their interconnection with varying levels up to the European Union (see Figure 2). Exemplary action points in such a framework could include "blueprint[s] for pioneering feasible regional energy initiatives". In contrast, regional sustainable development goals need to be integrated into national energy transition policy [35].

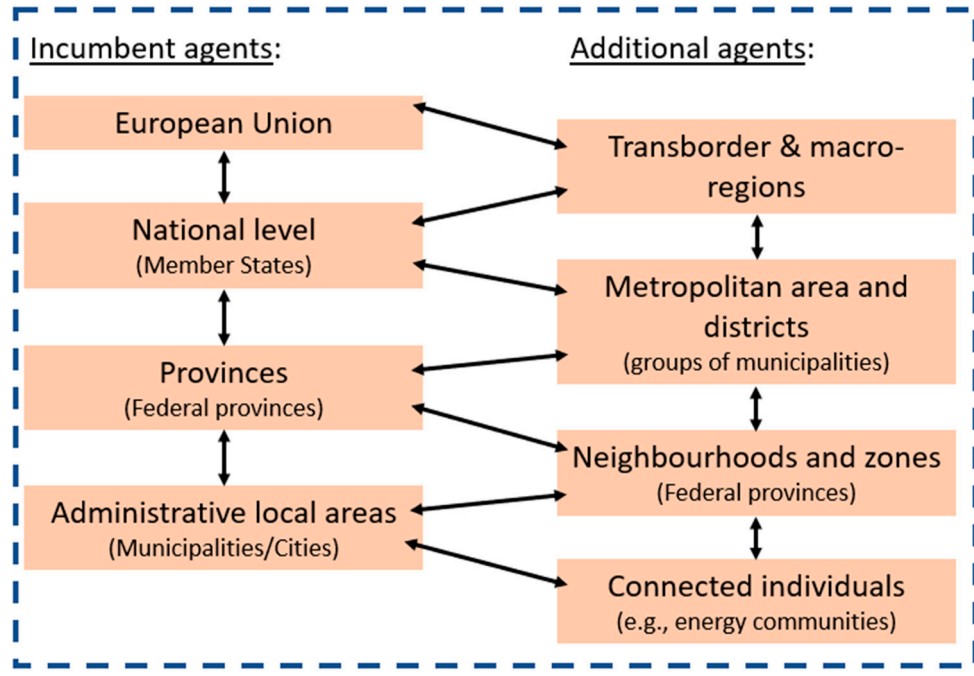

**Figure 2.** New integrated action space for multilevel governance. Source: own illustration modified from Dobravec et al., 2021 [34].

Renewable local energy initiatives historically focused on tackling problems related to social acceptance, such as the "nimby" (not in my backyard)-phenomenon. Today, especially modularity of renewable electricity generation, prosumer frameworks and demand-side management are rather coined by questions on social participation instead of acceptance. Participatory processes in governance and investments and providing energy production and consumption flexibility or engaging and nudging social networks to enhance energy efficiency or more sustainable consumption hold significant potentials, not yet recognized, e.g., in energy system planning [36–38].

The Renewable Energy Directive acknowledges the manifold "opportunities for growth and employment that investments in the regional and local energy production from renewable sources bring". The regional and local development opportunities include "export prospects, social cohesion and employment opportunities, particularly SMEs and independent energy producers", with decentralization fostering "community development and cohesion by providing income sources and creating jobs locally." The Renewable Energy Directive and the European Economic and Social Committee (EESC) mainly address "the role of civil society in the implementation of [decentralized P.V.- and wind energy]" [39]. Based on the findings of this paper, we think it is time to extend this broader socioeconomic benefits discussion to regional biomass mobilization. For example, a civic power plant ("Bürger* innenkraftwerk") based on P.V. already boosts the possibilities for participation manifold compared to a fossil-based power plant based on energy carrier imports. However, a local CHP-plant connected to a district heating network and supplied by forestry residues from forests primarily cultivated for stem-wood for wood construction and engineered wood products must exhibit an even higher societal participation potential.

Socioeconomic benefits of regional bioeconomies such as income, employment and net-profit for and of the engaged stakeholders are vital parameters to highlight here for policymakers and society (see, e.g., Wang et al. [40]). However,, the "inclusion of unique types of possibilities that each town or location offers" [41] needs to be taken into account, even though this might be more difficult to assess quantitatively. Furthermore, different levels of purchasing power parity (PPP) result in trade between regions [42]. While biomass export can provide economic benefits, the availability of ecologically sustainable options for meeting the own demand of the exporting regions has to be ensured. More indirect societal benefits include "protection against unpredictable energy pricing, improved energy access and security, reduction of transmission and distribution costs, independence from multinational utility interests and strategies and increased feasibility of renewable energy deployment within the framework of decentralized business cases" EESC in McGovern and Klenke [35].

For the present paper, we can outline that the valorization of the outlined socioeconomic benefits is still in its infancy. This observation is based on the fact that the quantification and assessments of the discussed aspects do not even take place in the theoretical energy system- and bioeconomy models today. Following Krumm et al. [37], we urge modelers to take "heterogeneity of actors, public acceptance and opposition, public participation and ownership" into account to at least theoretically explore the social dimension and benefits of regional biomass mobilization quantitatively.

### 3.2. Mobilization through Technological Innovation

Biomass mobilization will most likely be enhanced through various innovation types; organizational/institutional and social innovations can extend technological innovation. Organizational/institutional innovations address "changes in and among various organizational aspects of functions [of an organization or institution]", e.g., "the idea of networks—involving actors inside and outside [of the organization]" [43]. Social innovation is defined as "preferences of consumers, citizens, and workers for the types of products, services, environmental quality, leisure activities, and work they want" as well as respective changes in their behavior and interactions [43]. It also refers to "new solutions that imply conceptual, process, product, or organizational change, which ultimately aim to improve the welfare and wellbeing of individuals and communities" [44].

Still, and for this assessment level, we first and foremost focus on the market introduction and diffusion of technological innovations. We explore the current frontiers in adopting respective technologies for mobilizing the feedstock base of the circular bioeconomy. Especially biomass pre-treatment technologies (Sections 3.2.1 and 3.2.2), improvements in planning and harvesting (Section 3.2.3) and biomass production (Section 3.2.4) can provide opportunities for the mobilization of local, low-value and heterogenous biomass.

### 3.2.1. Decentralized Pre-Treatment

Classical, mechanical pre-treatment processes include chipping, pelletization, briquetting and bailing of biomass, which reduces transport and handling costs and better facilitates the storage and trade of densified bioenergy carriers [45]. Torrefaction, a mild form of pyrolysis, can further enhance relevant properties of the bioenergy carriers such as energy density, grindability and hydrophobicity [46]. Pyrolysis to maximize the liquid fraction of the output [47], hydrothermal treatment [48,49], upgrading of biogas from anaerobic digestion [50] or from biomass gasification to biomethane [51] are other strategies to facilitate biomass mobilization. A large number of publications focused on the impact of densification technologies on decreasing supply chain costs [52] to promote biomass commodification and trade [53] and to improve conversion efficiency, for example, through gasification [54].

In contrast to properties related to trade also the value of energy and carbon being reliably stored over a long time becomes particularly relevant in a Circular Bieoconomy [45]. This value is based on improved volumetric energy densities and suitability for storage in existing infrastructures in light of flexibility needs due to increasing shares of intermittent renewable energy production. However, similar to the socioeconomic benefits outlined in Section 3.1, tools to quantify and valorize this flexibility, even in theoretical energy system models, are missing today (see, e.g., Thrän et al. [55]).

### 3.2.2. Mobile/Portable Pre-Treatment

Investments in conversion- and pre-treatment plants are primarily driven by economies of (unit-) scale. It is recommended to optimize between plant size and the "respective feedstock supply distances for various feedstocks, supply modes and feedstock yield, availability and accessibility combinations" [45]. This obvious connection has far-reaching consequences, including the need for pre-treatment steps based on commodity markets (see Section 3.3), emerging overcapacities with increasing feedstock competition and tendencies to create vertically integrated supply chains.

Still, some niche actors aim to down-scale respective stationary technologies for them to become relocatable, transportable or even mobile. Polagye et al. [56] outline in a detailed cost comparison how the economy of scale results in the issue that "the production of bio-fuels using mobile and transportable facilities is significantly more costly than production at a stationary or relocatable facility." A more recent project, "mobileflip.eu" from VTT and SLU, acknowledges that the added value of the smallest functional unit would be reflected by its flexibility to switch between feedstocks that are scattered spatially but also in time. They discuss mobile pre-treatment facilities of 687 tonnes of forest residues input per year [57]. De la Fuente et al. [58] outline the LCA of mobile pelletization, torrefaction, slow pyrolysis, hydrothermal pre-treatment and carbonization and respective environmental challenges for downscaling pre-treatment. Demonstration projects are furthermore described in Mirkouei et al. [59,60] for mobile bio-oil production (i.e., the Renewable Oil International LCC) based on a relatively old refinery concept from Badger and Fransham, 2005 [61] and another slow pyrolysis for biochar production (Schatz Energy Research Center) as an alternative to slash pile burning [62,63]. Some commercialized concepts exist, such as the relocatable shipping container "PelletBox" by Prodesa and mobile pelletization plants such as the "Krone Premos 5000', the "Schaider Groups Pelletec", the "Gmco mobile pellet plant" and the "Proxipel concept". However, while these concepts could significantly help mobilize local and heterogenous biomass resources, they do not play a considerable role in current provision structures.

### 3.2.3. GIS Supported Planning and Harvesting

Subramanian et al. [64] classify energy system models regarding their (1) decision-making hierarchy (strategic, tactical, operational) and (2) the level of technology aggregation (unit operation, plant, supply chain, energy sector and whole economy). However, we can observe supply chain innovation and the utilization of Geographic Information

Systems (GIS) on all three hierarchy levels and imagine supply chain management to become the backbone of connecting organizational planning to mobilize biomass for a circular bioeconomy.

However, a notable literature review on biomass supply chain optimization by Ba et al. [65] finds that "GIS [is] mainly used in a strategic context because they lack the short and medium-term temporal dimension that is required for tactical and operational decisions." They are often applied to find economically optimal solutions for mobilizing forestry residues. In contrast to forestry biomass, optimization of agricultural residues and/or based on environmental or social parameters (or a combination thereof) are discussed to be less plentiful [66]. Economic optimization addresses minimizing costs for public endeavors and society, e.g., drafting policy recommendations or maximizing revenues for private projects and investment decisions. The "BeWhere model" (based on Leduc, [67]), for example, "identifies the localization, size and technology of the renewable energy system that should be applied in a specific region". At the same time, Frombo et al. in Ba et al. [65] provide a "GIS-based Environmental Decision Support System (EDSS)" to support investment decisions for location and size of pre-treatment plants for forest residues.

The utilization of GIS data in a strategic context includes decision variables such as the optimal location to construct a biomass densification or conversion facility, its capacity, technological set-up and the biomass supply and distribution between facilities and the end-user. In contrast, tactical models aim at inventory planning and identify optimalharvest quantity, harvest schedules, inventory deployment and optimize transport modes, shipment size and routing. These models are mainly business-oriented, such for example, the "VITO MooV model" (based on De Meyer et al. [68]) optimizing supply chain configurations, including decisions on transport mode planning, storage capacity planning and feedstock- and product variability for medium-term (i.e., next months to years) scheduling. Vopenka et al. [69] furthermore describe a tool for spatial and temporal optimization of forest harvesting in a user-friendly digital map, potentially bridging the gap between tactical and operational planning.

The operational context can be discussed regarding scheduling activities in a temporal granularity below months and weeks. An H [70] aims to optimize the "daily scheduling of [trucks and] mobile [loaders] to transport biomass from satellite storage locations to a bioenergy plant" and present a case study on corn stover. Zamar et al. [71] identify the best daily routing schedule for trucks to collect sawmill residues for energy conversion in the pulp and paper industry. Besides these rather classical travelling salesman problems, the literature on biomass supply chain GIS-modeling for operational decision support is scarce.

The view citations mentioned in the central reviews on biomass supply chain modelling [65,68,72,73] can mostly and arguable be better grouped in the tactical or even strategic context. We acknowledge that thorough market research would be more thankful for this type of model than the scientific literature research performed for the objective of this paper. Innovations in the field of precision agriculture, including optimizing fertilizer- and pesticide application, as well as harvest scheduling, weather forecasts of high temporal and spatial resolutions but also dynamic record-keeping based on data collection from satellites, drones and on the ground (e.g., https://geomarvel.com/, accessed on 31 December 2021), can be used for increasing the mobilization of biogenic residues. Digitally guided forest management, planning for collecting and utilizing damaged wood from extreme weather events or minimizing soil contamination through harvesting after natural washing (rain) and optimized deployment of mobile pre-treatment could be potential applications. These big data strategies can be complemented by further digitalization and mechanization efforts, e.g., in silviculture operations and with soil mechanics fundamentals to assess terrain trafficability as, e.g., currently developed in the "H2020 EFFORTE project".

### 3.2.4. Next-Generation Primary Sources

Wild cards in the bioenergy and the circular bioeconomy discussions can be seen in novel biogenic carbon sources and -sinks for biogenic carbon. Their particular potentials

for regional mobilization strategies supporting a circular bioeconomy are challenging to discuss. Alternatively, we provide a short overview of the different research frontiers that will have to be assessed in follow-up projects in detail.

The production of short rotation coppice (SRC) remains relatively low (see Section 3.1.1). Various projects assess the potential of crops such as Miscanthus and hemp, such as the "BBI GRACE", or agricultural prunings and plantation removal in the "uP_Running project". The EU-Brazil cooperation in the "BECOOL project" addresses different "annual and perennial dedicated lignocellulosic crops, together with crop and process residues such as cereal straw, sugar cane straw, bagasse, and lignin-rich residues." The "MAGIC" and the "ADVANCEFUEL" projects have an even broader scope for abundant oil, lignocellulosic, carbohydrate or specialty crops. Innovation for agricultural management such as optimization of planting density, crop establishment improvements, crop rotation intercropping, multi-purpose cropping, cropping on marginal land and precision farming, and increasing the harvesting frequency, will potentially provide additional biomass. These topics are subject, e.g., to the "LIBBIO" project, the "SeemLA" project and the "FORBIO" project.

In addition, "technological advances in agriculture and forestry can still be expected through improved fertilization, breeding, crop selection, and gene editing, and genetically modified organisms not only for yield improvements but also to provide resilience against temperature tipping points [for biomes] caused by global warming" (see Duffy et al. [74]).

In this line, the production of micro-and macro-algae also have to be mentioned. Algae are "produced in photobioreactors, in open ponds or harvested from the natural environment are also promising primary feedstocks and should be addressed, e.g., in bioeconomy modelling and discussions" [75]. As part of the "blue economy", this is mainly commercially realized for food or specialty food products (e.g., Omega-3 fatty acids). The high water content renders energy or chemicals production, particularly energy and cost-intensive. Furthermore, services such as nutrient recycling and recirculation or urban solutions are still in their infants, often discussed under the umbrella of Nature-Based Solutions [76].

### 3.2.5. Next-Generation Primary Sinks

The IEA Bioenergy project on the deployment of biocarbon capturing and sequestration published three case studies of large scale BECCS coupled with CHP in Denmark [77] for bioelectricity only in the Drax Power Station in the U.K. [78] and with the waste incineration plant of Fortum Oslo Varme in Norway [79]. In addition to the "centralized" and large-scale BECCS, more decentralized carbon storage solutions, such as halting deforestation and degradation, have the most significant carbon emissions mitigation potential followed by afforestation (non-forest areas to forests) as outlined, e.g., in [80], reforestation (deforested areas to forests) (see Chazdon et al. [81]) and forest restoration (degrading forest to healthy forest) [82]. Furthermore, Fritsche et al. [83] lists biochar addition to soil "improving water holding capacity and nutrient use efficiency" while sequestering carbon. Especially in the light of climate change, we want to stress that these decentralized carbon management strategies will require substantial efforts attracting skilled labor to rural areas and providing mobilization opportunities for circular bioeconomy feedstock.

### 3.3. Market Creation for Biomass Mobilization

Zooming into the regional context of biomass mobilization, we find that existing legislative frameworks (Section 3.1), readily deployable technologies, and niche innovations (Section 3.2) are often pre-conditions to establish economic activities but do not necessarily result in such. The creation and establishment of dedicated and functioning physical markets, regional-, interregional- and international trade depend on additional factors, such as market competitiveness and -liquidity [83]. In the following pages, we provide and discuss selected strategies on the market creation level for the mobilization of low-value and heterogenous biomass feedstock.

### 3.3.1. Market Catalysts for Wastes, Residues, Post-Consumer Products and Secondary Raw Materials

Regarding European efforts to transform the economy, the European Bioeconomy Strategy states that "to be successful, the European bioeconomy needs to have sustainability and circularity at its heart" [84]. A comprehensive European policy overview on circularity measures can be found in Milios L. [85] who outlines, that even though the E.U. can be seen as a leader in circularity, its actions mainly focus on the end-of-life phase of consumables so far while avoiding waste through improved quality and repair options are rather novel concepts. Waste collection, processing and treatment gained a high priority in the many E.U. Member States, distinguished by waste fractions, e.g., containing fossil-based plastic, biogenic waste and electronic waste. The E.U. Circular Economy action plan [2] thus goes a step further by aiming to create a secondary feedstock market, primarily focused on fossil-based plastics from waste collection but also including residues from downstream industries. This creates market opportunities for commercial waste and residues plastic collectors and distributors (e.g., see https://polymerstocklist.com/, accessed on 31 December 2021). The collection and mainly energy utilization of waste wood, also coined post-consumer wood, is already better established, resulting in small but quantifiable international trade [7]. Similarly, the collection of used cooking oil and animal fats for international trade and biodiesel production has gained importance in the recent decade [86]. While used cooking oil and animal fats exhibit a limited potential for business creation, post-consumer wood from traditional applications but also from increased utilization, e.g., in a potentially growing wood-based construction sector (see, e.g., Churkina et al. [87]), as well as novel materials such as biopolymers (see, e.g., Schipfer et al. [14]), will demand residues and waste collection, processing, treatment, intermediary storage and distribution to biobased industries.

### 3.3.2. Physical and Virtual Bio-Hubs

Decentral or regional biomass processing depots, or bio-hubs, are facilities that are discussed to overcome the mismatch between the distributed occurrence of biogenic resources and large-scale centralized conversion plants such as biorefineries [88–90]. Conceptually, by including on-site pre-processing and/or densification technologies, bio-hubs are enabling regional market creation to allow farmers and forest owners to convert their residues into valuable by- or even co-products in the form of bioenergy or biogenic carbon carriers. Residue collection and forwarding can be either done by the farmers themselves or third parties eventually owning mobile pre-treatment equipment (see Section 3.2) or machinery to make the residues accessible. Economically feasible options for residue collection are often highly case sensitive, mainly based on low energy densities and high water content and thus limit economically viable transportation ranges [45]. Different collection options exist and should be compared on a case-by-case basis.

A dedicated project for collecting the best bio-hubs practices was initiated within the IEA Bioenergy TCP. The IEA Bioenergy Biohub project collects and disseminates case studies from different world regions and theoretical considerations regarding agri- and silvicultural residues collection centers (see https://arcg.is/qLqaK, accessed on 31 December 2021). The broad definition of bio-hubs by Nasso et al. [91] valorize opportunities such as "streamlining of processing, storage and transportation, reduction in administrative costs, making a variety of biomass types available at a single location, providing an opportunity for suppliers of biomass products to continue producing in the offseason [ . . . ], as well as a place for companies to connect and trade with one another" is used for the projects' assessment. Successful examples include the Tschiggerl Agrar GmbH, a logistics center for processing agricultural residues to feed, animal bedding material and fuel, but also virtual bio-hubs such as the Rosewood Network for knowledge transfer in eastern European countries, discussed in the IEA Bioenergy Biohub workshops [92].

Other virtual bio-hubs have also been initiated several times throughout history. While some aim to facilitate knowledge transfer (see Table 1), others establish online

market platforms to trade bioenergy commodities. Projects such as "b2bbioenergy.eu" and "promobio.eu", "pellet-zone.com" and "bioexchange.com" can be named within this realm. While these projects did not succeed and homepages are not maintained anymore, the "Biomass Commodity Exchange (BCEX)" for cellulose biomass trade within the U.S., including energy crops such as willow, poplar and switchgrass, is under construction. For more international trading of biomass and bioenergy commodities, the wood pellets futures from "Euronext" and platforms to prove the origin, especially of liquid and gaseous commodities such as the "German Nabisy" system or national and international renewable gas registries fall in the category of virtual bio-hubs. In the Baltic States, "Baltpool" enables the trade of biomass feedstocks and even heat, with the latter being reported a relatively underdeveloped market in other parts of the E.U. [93,94]. If virtual market platforms can alter regional trade remains to be shown. However, the comparison, implementation, development, and impact of virtual bio-hubs could be an exciting field of energy economic research without considerable publications to date.

Similarly, virtual bio-hubs include the "Electronic Reverse Auction (eRA)" system in Denmark. Via the eRA straw-based thermal power stations source their fuel since 2006. A power station initiates the auctioning by requesting a certain amount and quality of straw. Various potential suppliers can underquote (reverse auctioning) until a determined deadline. Each buyer runs its auctioning model, while the supplier can decide on amounts, contract periods (1–3 years) and prices based on blind auctioning.

The applicability of eRA for the German straw market has been discussed in Pfeiffer et al. [95]. Even though the eRA system is a virtual market platform, it still requires intensive relationship management. This results in additional work for the buyer and significant competition among the sellers. Furthermore, it has to be mentioned that eRA in Denmark mainly was implemented to stabilize a market, not to establish it. For new markets, feasibility studies need to address supply and demand potential, engagement of possible stake- and shareholders and include a thorough risk assessment [95].

### 3.3.3. Commoditization of Intermediary Products

In the previous years, the IEA Bioenergy TCP Task40 discussed the commodification of bioenergy carriers as a necessary step for market uptake and market creation. For example, in Olsson et al. [83], we present the key features of commodity markets, including trade with standardized and perfectly fungible (interchangeable) units. These product-related properties are complemented with market-related properties, such as the engagement of many buyers and sellers, resulting in high market liquidity. These properties can be expressed by international trade and equilibrating forces on the last remaining product differentiator, the product price. In Schipfer et al. [96], we try to quantify the commoditization of wood pellets based on competitive spatial equilibrium modelling using modern trade theory. The wood pellets market can be seen as a role model for possibly upcoming pretreated bioenergy carriers such as straw pellets, briquettes, pyrolysis oil and biomethane and biogenic carbon-based (liquid) chemicals, including biodiesel, bioethanol and liquid organic hydrogen carriers (LOHC). Still, we find a relatively inefficient international wood pellets market in central Europe with high margins for traders. The identified low efficiency and low commoditization could result from low market transparency and an "intrinsic valuation of non-quality related properties like pellets color and more regional biomass supply chains". In the article we argue, that internationally harmonized data and sustainability standardization (and awareness thereof) could steer the market towards a joint price benchmark and more stable and on average lower prices. However, we acknowledge, that this development holds the risk to drive smaller and less economic viable producers with regional supply chains out of the market.

A more critical view and direct reaction on this topic are presented in McGovern and Klenke [35]; here, commoditization is deemed "counter to generating viable regional energy projects as it reduces the stakeholder role of local agricultural biomass producers". Commoditization is addressed as a challenge with the potential impact to "seriously decel-

erate or weaken the movement towards energy democracy and decentralized renewable energy". Provided supporting arguments include that longer and international supply chains give citizens "little-to-no influence over the sustainability of their energy supply" and pose a risk of distorting ecological equilibrium since they undermine efforts for (local) circularity. This critique understands the heterogeneity of regional biomass supply chains as an essential driver for the participation of diverse stakeholders, providing ownership, local revenue and job creation, and independence from multinational utility interests.

Some of these commoditization-critical arguments are not sufficiently substantiated; e.g., sustainability certification provides control over the energy supply chain, even for internationally purchased wood pellets, wood pellet supply chains, especially for residential heating, are by no means centralized. However, the overall controversy and potential trade-off between commoditization and specialization, hence between larger and efficient markets and smaller and less efficient markets, is highly relevant, especially in the light of regional and civic energy initiatives. As discussed under the legislative strategies (see Section 3.1), circular bioeconomic structures have exceptionally high decentralization- and societal participation potentials. We propose that this key performance indicator (KPI) could be better measured by the number of stake- and shareholders part of a specific bioeconomy supply chain rather than by the transport distance of bioenergy or biogenic carbon carriers. Still, the number of stake- and shareholders, similar to the transport distance, contributes negatively (higher costs) to the accumulated costs of the final product/service and for the consumer. Without "giving credits" to this KPI, commoditization and the resulting (international) market efficiency indeed result in (1) less stake-/shareholder diverse supply chains and (2) relocation of supply chains to regions of lower production costs. In addition, and in line with Section 3.1, biomass export from lower PPP-regions does only provide economic and ecological benefits, as long as it does not endanger the availability of sustainable options to meet the demand of the exporting regions.

Sustainability certification can ensure that efficient markets do not race to the lowest environmental standards and even elevate standards and transfer higher standards to some regional markets. Additional socioeconomic certification for smaller producers, as implemented for some products (fair-trade of coffee and tea, chocolate), can be understood as the specialization of the supply chain; however, a critical mass of equitable supply chains could set new standards for commodity markets. Extending sustainability certification of internationally traded biomass commodities with socioeconomic KPIs such as stake-/shareholder variety could be a promising strategy for just and regional biomass mobilization. Research on comprehensive frameworks for sustainable assessments of biobased products is performed, e.g., in the Horizon2020 STAR-ProBio project.

### 3.3.4. Informative Networks for Knowledge Exchange and Market Creation

Transparency is critical for functioning international markets and is also expected to play an essential role in regional markets to support values (e.g., sustainability, stake-/shareholder diversity) but also stability through the exchange of harmonized price information, spatially and temporally explicit production and consumption data and data on flows and stocks, including storage volumes. Bioenergy storage, and its flexibility service character, is of particular importance when energy systems develop towards higher implementation of intermittent renewable energies such as photo-voltaic and wind power. Market transparency induces fair competition, which can be beneficial not only for the end-users but also for the shareholders of the supply chain if the information on innovation is transparently traded and best practices are shared. Currently, market information networks are collecting, preparing and providing respective knowhow and knowledge, often in cooperation with experts from industry and academia:

The IEA Bioenergy TCP and related TCPs (see Figure 3) is part of the intergovernmental OECD/IEA network. It is committed to providing scientific backed information on the level of markets, developed- and immature technologies and how their status today, opportunities and barriers for market diffusion in the upcoming decades.

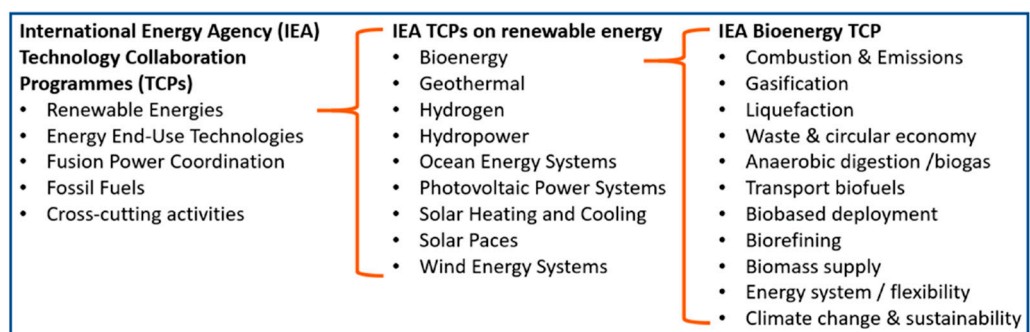

**Figure 3.** Structure of the IEA network and technology collaboration program. Source: own illustration based on BMK, KG and Eggler, Indinger and Zwieb, 2018 [96].

Other energy-related international and partly intergovernmental information networks include the International Renewable Energy Agency (IRENA), the U.N. initiative Sustainable Energy for All (SE4ALL), the REN21 and many more (e.g., International Energy Forum /IEF, Global Bioenergy Partnership/GBEP, BioFuture Platform, EurObserv'ER, European Renewable Energy Council/EREC, World Energy Council and Food and Agriculture Organisation on bioenergy and food security/FAO BEFS). On a European level, especially the European Energy Research Alliance (EERA) has to be mentioned. While these networks focus on the exchange between countries, also considerable efforts are undertaken for inter-regional exchange of know-how and knowledge. Networks on an inter-regional and regional level often include national and regional governments and regional energy agencies, and interest groups. They are supported by international or national structural and regional development funds (see Section 3.1). Analyzing the unique selling propositions and the effectiveness of the different networks and initiatives is beyond the scope of this project. Still, it could be an exciting research topic for participatory research and research on international consortia, potentially resulting in harmonized performance metrics and ultimately improved impact on regional sustainable energy implementation.

## 4. Conclusions

We assess mobilization strategies for local, low-value and heterogenous biomass feedstock. Respective feedstocks, including energy crops-, forestry- and agricultural residues and biogenic wastes, are identified as the backbone of the circular bioeconomy of tomorrow. In contrast to currently deployed biomass for energy purposes, novel provision structures face considerable technical-, societal- and organizational challenges.

To explore and discuss these challenges, we are building upon the IEA Bioenergy Task40 expertise on international bioenergy trade and the current provision of bioenergy. For the present paper, we aim at transferring and extending our knowledge on current bioenergy carrier provision structures to the local, low-value feedstock base of the circular bioeconomy.

This approach exhibits limitations on each of the three assessment levels: The (1) legislative framework level limits the scope to top-down frameworks and the E.U. policy landscape. A first attempt to overcome the top-down view is made by bringing together findings from international projects on regional biomass mobilization. The (2) innovation level assessment is limited to technological innovation only. This limitation is mainly due to the available expertise in the consortium. However, social and organizational innovations also visibly coined the findings of all three assessment levels. For the (3) market creation level, the results and discussion section builds on the authors' particularly strong scientific background. The remaining limitations concern the under-researched nature of this area; they include limited scientific, energy-economic publications on electronic bioenergy carrier trading or comparative discussions of different market structures and market instruments.

We find that the E.U. policy landscape, especially under Covid recovery's umbrella, provides significant funds for regional development and biomass mobilization. Most

regional action plans shifted to international, quantitative potential assessment approaches to provide biomass to different Bioeconomy sectors. The next frontier can be seen moving towards Multilevel governance, entangling governance levels from neighborhoods to E.U. governance. Respective opportunities for regional biomass mobilization are particularly exciting since the circular bioeconomy exhibits an outstanding decentralization- and thus resource democratization potential.

The niche technological innovations for the mobilization of local, low-value and heterogenous biomass resources already exist. They include mainly pre-treatment technologies, GIS-supported planning and novel primary biomass sources. However, ensuring economic and ecological sustainability while down-scaling pre-treatment technologies to the smallest possible functional unit need to be addressed. In return, mobile or at least portable pre-treatment and densification plants could induce valuable operational flexibility. Mobile pre-treatment, coupled with big-data and GIS support, could overcome the challenges of seasonal fluctuations and in-homogenous geographical feedstock distribution.

However, markets for local, low-value and heterogenous biomass resources are largely underdeveloped. Physical- and virtual bio-hubs and market platforms are required to connect the highly diverse supply side with the demand side for biomaterials and bioenergy. Today, these hubs are rather an exception than the rule; numerous attempts of establishing virtual market platforms have failed over the years. The heterogeneity of market actors and the traded goods can be identified as a major challenge, also for successful platforms such as the eRA-straw market. With this regard, commoditization is addressed as a double-edged sword; without environmental and socioeconomic standards, market creation might be at the expense of biodiversity and stakeholder variety.

The challenges and opportunities of the three assessment levels point towards a common denominator: The quantification of the systemic value of strengthening the potentially last remaining primary economic sectors, forestry, agriculture and aquaculture, is missing. With the eroding importance of other primary economic sectors, including fossil fuel extraction and minerals mining, the time is now to assess and act upon the value of the supply side of a circular bioeconomy. This value includes the support the Bioeconomy can provide to structurally vulnerable regions by creating meaningful jobs and activities in and strengthening the resource democratic significance of rural areas.

Energy system and circular bioeconomy modelling could play an important role, theoretically simulating the systemic value, e.g., of temporal- and spatial flexibility of pre-treatment technologies and of stakeholder diversity in markets and multilevel governance. Therefore, modelling should account for multiple assessment criteria and modelling functions, based on all types of resources, including monetary-, natural-, CO2-budget- but also human resources. Based on the theoretical work, sound recommendations for biomass mobilization action plans, technology investment decisions and market organization should be derived.

**Author Contributions:** Conceptualization, F.S. and A.P.; methodology, F.S. and A.P.; investigation, F.S. and A.P.; writing—original draft preparation, F.S.; writing—review and editing, F.S. and R.H.; project administration, F.S.; funding acquisition, F.S. and R.H. All authors have read and agreed to the published version of the manuscript.

**Funding:** This research was funded by the IEA Bioenergy TCP Task40 and the BMK, FFG grant number 876720.

**Institutional Review Board Statement:** Not applicable.

**Informed Consent Statement:** Not applicable.

**Data Availability Statement:** Not applicable.

**Acknowledgments:** We want to warmly thank the three reviewers and the editor for the valuable comments and the fast but thorough reviewing process. Your input improved this paper considerably. We also want to thank the Open Access Funding by TU Wien library.

**Conflicts of Interest:** The authors declare no conflict of interest. The funders had no role in the design of the study; in the collection, analyses, or interpretation of data; in the writing of the manuscript, or in the decision to publish the results.

## Nomenclature

| | |
|---|---|
| BBI JU | Biobased Industries Joint Undertaking |
| BECCS | Bioenergy Carbon Capture and Sequestration |
| CAP | Common Agriculture Policy |
| CBE JU | Circular Bioeconomy Joint Undertaking |
| C.E. | Circular Economy |
| CHP | Combined heat and power |
| CO2 | Carbon dioxide |
| EESC | European Economic and Social Committee |
| EFA | Ecological Focus Areas |
| eRA | Electronic Reverse Auctioning |
| E.U. | European Union |
| GHG | Greenhouse Gas |
| GIS | Geographic Information System |
| IEA | International Energy Agency |
| KPI | Key performance indicator |
| M.S. | Member States |
| MLG | Multilevel governance |
| MWel | Megawatt electric |
| NUTS | Nomenclature of Territorial Units for Statistics |
| OECD | Organization for Economic Co-operation and Development |
| P.V. | Photo voltaic |
| R&D&D | Research, development and deployment |
| SME | Small and medium enterprises |
| TCP | Technology Collaboration Programme |
| U.K. | United Kingdom |
| U.S. | United States |

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
