# Peer review of "Strategies for the Mobilization and Deployment of Local Low-Value, Heterogeneous Biomass Resources for a Circular Bioeconomy"

_energies, doi:10.3390/en15020433_

Round 1

Reviewer 1 Report

Dear Authors,

The topic is interesting and up to date.  However, some improvements are needed to make the manuscript in line with journal publication standards.

1. Introduction:

While the authors establish some links to some extant literature, authors need to establish a more coherent framework for the overall paper. That means, the introduction should clearly indicate the need for this paper in relation to extant research studies.
- Some methodological insights should be added to the Introduction.
- Please review the literature gap analysis, assuring an internal coherence between research topic,
emerging findings, knowledge gaps and research goals (and then, research questions).

Outline the CE concept more broadly and justify the need for publication more clearly. 

It is worth referring to the following publications: 

Fonseca, L.M.; Domingues, J.P.; Pereira, M.T.; Martins, F.F.; Zimon, D. Assessment of Circular Economy within Portuguese Organizations. Sustainability 2018, 10, 2521. https://doi.org/10.3390/su10072521 

Shih, D.-H.; Lu, C.-M.; Lee, C.-H.; Parng, Y.-J.M.; Wu, K.-J.; Tseng, M.-L. A Strategic Knowledge Management Approach to Circular Agribusiness. Sustainability 2018, 10, 2389. https://doi.org/10.3390/su10072389 

etc.

3. Methods: Please discuss in detail how you analyzed the data in order to present the key findings.

4. Develop a separate "discussion"  and "Conclusions" section.  The findings are a good basis for discussion but authors have to answer the following questions: What does this research tell us that we didn’t already know? What new does this paper bring to the table? This paper will have more value if the authors can tie the conclusions with numerical results from their study and compare them with the findings by previous researchers.

5. Conclusions are very weak and almost not present. the Implications for research, theory, practice and society are not clear though I can see that these aspects can be elaborated further. Research limitations must be explained.

Good Luck !

Author Response

Dear Reviewer

Thank you for your valuable comments and input and especially for your reviewing time. The manuscript has been improved considerably based on your feedback and the feedback from the other two reviewers and the editor.

Please find replies to your specific comments attached.

Kind regards
Fabian Schipfer

Reviewer 2 Report

Dear authors,

Thank you for giving me a good read. The paper is already in a good quality, rarely seen lately. Yet, there are some minor items that need to be improved before publishing.

There is only three concerns of substance, whereas the rest of the comments are mostly cosmetics. 

Details are in the document attached.

Best regrads,

Author Response

Dear reviewer

Thank you very much for your valuable time and review. Please find attached the reply to your comments and inputs.

Best regards

Reviewer 3 Report

The work is complete, detailed and provides a 360-degree view of the current situation and development possibilities of the biomass bio-economy,

However, I believe that a general rereading of the paper is necessary to make it more fluent: reading the paper is difficult, many citations are inserted without being introduced first (at least the background) and sometimes the sentences are too long and with too many subordinates. The creation of paragraphs, for example, could help.

Here are some notes on the manuscript:

73 —> 92 —> I think the concepts need to be rearranged. There is a lot of random information, but a complete narrative is missing.

134 —> 140 : this paragraph, in my opinion, should be moved to the introduction

Paragraph 3.1: I think this paragraph fitting better in Material and Methods section.

170: What exceptions?

277 —> 284: it would be interesting to go into detail on this issue. So far you have talked about investments and projects on financing the bio-economy. You then said that the funds for the production of energy-only are gradually being reduced. Therefore, investigating what and how future investments in this field could be, would increase the quality of the paper.

337 —> new paragraph

394 —> 395: add if the “mobile pre-treatement” is economically sustainable and/or if it is feasible only for a quantity of material above a certain threshold.  What size of facilities are we talking about?

481 —> “ bioms’ “ is this a typo error?

482 —> new paragraph for algae.

581 —> 594: re-write the paragraph. A first time reading is difficult and I think some information is missing to understand the example reported.

The conclusions are very clear and fully summarize the work. I would only propose two changes that make the paper more readable.

724 —> 725: adding a comment, from experts on the subject, on what could be the future of these hubs: physical, virtual, hybrid? Which solution would boost the system to develop it in the short term?

737 :  explain better the concept of “systemic value of strengthening knowledge-based primary economic sectors”.

Author Response

Dear reviewer

Thank you for reviewing this work and your valuable comments and inputs. Attached we explain in detail, how your input improved the manuscript.

Best regards
Fabian Schipfer

Round 2

Reviewer 1 Report

I have no more comments.

Good luck!

Reviewer 3 Report

Thanks to all the Authors for the corrections.

best regards